# Symblepharon, Ankyloblepharon, and Salt Gland Dysfunction in a Loggerhead Sea Turtle (*Caretta caretta*)

**DOI:** 10.3390/vetsci9060281

**Published:** 2022-06-08

**Authors:** Andrea Affuso, Cristina Di Palma, Leonardo Meomartino, Antonino Pace, Serena Montagnaro, Valeria Russo, Giuseppina Mennonna, Fabiana Micieli, Fulvio Maffucci, Sandra Hochscheid, Francesco Lamagna, Ilaria D’Aquino, Barbara Lamagna

**Affiliations:** 1Marine Turtle Research Group, Department of Marine Animal Conservation and Public Engagement, Stazione Zoologica Anton Dohrn, 80055 Portici, Italy; andrea.affuso@szn.it (A.A.); antonino.pace@szn.it (A.P.); fulvio.maffucci@szn.it (F.M.); sandra.hochscheid@szn.it (S.H.); 2Department of Veterinary Medicine and Animal Production, University of Naples “Federico II”, 80137 Naples, Italy; serena.montagnaro@unina.it (S.M.); valeria.russo@unina.it (V.R.); fabiana.micieli@unina.it (F.M.); lamagna@unina.it (F.L.); ilaria.daquino@unina.it (I.D.); blamagna@unina.it (B.L.); 3Interdepartmental Radiology Centre, University of Naples “Federico II”, 80137 Naples, Italy; meomarti@unina.it (L.M.); pina.mennonna@gmail.com (G.M.)

**Keywords:** sea turtle, loggerhead, symblepharon, ankyloblepharon, salt glands, dacryoliths

## Abstract

Adhesions involving the bulbar and the palpebral conjunctiva (Symblepharon) may interfere with tear drainage, cause chronic conjunctivitis, and reduce ocular motility. This condition may be associated with adhesion of the edges of the upper and lower eyelids (ankyloblepharon). The present case describes bilateral symblepharon, ankyloblepharon and salt gland dysfunction in a juvenile *Caretta caretta*. The loggerhead presented both eyelids swollen, ulcerated, and not separable when rescued. Eye examination was not possible, but ultrasonography showed right bulbar integrity, while the left eye was smaller, with a thicker cornea that had lost its normal doubled lined structure. Surgical dissection of the fibrous adhesions between the palpebral and bulbar conjunctiva, cornea, and third eyelid was performed, and large dacryoliths were removed. The microscopic findings were consistent with chronic keratoconjunctivitis. Ultrastructurally, no virus-like particles were observed. In addition, tissue samples were negative for herpesvirus by qualitative PCR. The eyelids of both eyes and the corneal epithelium of the right eye healed; moreover, the vision was restored in the right eye. There were no recurrences after 12 months of follow-up, and the turtle was released 16 months after the end of treatments on the southern Tyrrhenian coast in the western Mediterranean Sea. To the authors’ knowledge, this is the first report of symblepharon with ankyloblepharon and salt gland dysfunction in *Caretta caretta* turtle. Ocular ultrasonography was helpful in the preliminary diagnostic work-up.

## 1. Introduction

The most common ocular injuries in sea turtles are blepharitis, keratitis, and conjunctivitis; fibropapillomatosis, which maybe is caused by a virus in the alphaherpesvirus family, is also frequent [1,2]. An atypical case of conjunctivitis by ChAHV-5 was recently described in a captive loggerhead turtle [3].

The eye in the Loggerhead Sea turtle has three keratinized eyelids, two mobile, dorsal and ventral eyelids, and a nonmobile medial lid called the “secondary lid” [4,5]. The ventral lid and nictitating membrane are continuous with the conjunctiva. The salt gland is the largest organ in the head of sea turtles. There is also an Harderian gland located anterior to the eye. Each salt gland envelops the caudal orbit, extends caudally and medially to the eye, and drains into a duct that opens in the caudo-dorsal conjunctiva [4,5]. This organ enables sea turtles to maintain osmotic and ionic homeostasis in a hypersaline environmental medium by secreting a concentrated sodium chloride solution in response to increased plasma sodium [6,7,8].

Adhesions involving the bulbar and the palpebral conjunctiva may impair tear drainage, produce chronic conjunctivitis, and retard ocular motility, resulting in the disease called symblepharon [9]. This condition occurs when two epithelialized surfaces (one is the conjunctiva) ulcerate and the substantia propria of these two surfaces adhere to each other, involving the nictitating membrane and the cornea in many cases [10].

Symblepharon results from a variety of ocular traumas, keratitis, or conjunctivitis [11], and it can be associated with eyelids adhesion (ankyloblepharon), with variable severity and prognosis [12]. In veterinary medicine, symblepharon is commonly observed in cats as a result of feline herpes virus (FHV-1) infection, and although it can be corrected successfully, keratoconjunctival inflammation is often chronic, so symblepharon may recur [9]. Recently, an atypical case of conjunctivitis by ChAHV-5 was described in a captive loggerhead turtle with bilateral chemosis, conjunctival hyperemia, and mucoid secretions [3].

Symblepharon rarely occurs in dogs and is usually a result of trauma or chemical burns [13], although one case of symblepharon secondary to ophthalmomyiasis externa has been described in a 4-month-old crossbred female dog [14]. In horses, symblepharon may follow trauma [9]. Symblepharon has also been described in birds and is associated with conjunctivitis and, in a few cases, septicemia [15].

The present case describes a bilateral symblepharon associated with ankyloblepharon and salt gland dysfunction in a juvenile *C. caretta*.

## 2. Case Presentation

### 2.1. Clinical Findings

A juvenile loggerhead turtle (*C. caretta*) of unknown sex, measuring 59.2 cm standard curved carapace length (CCLst) and 20.6 kg body weight (BW), stranded on the Adriatic coast of Puglia (Italy), was rescued by the sea turtle rescue center “Natural History Museum of Salento-Calimera-Lecce”, where the turtle received first aid. Blood parameters were within the range indicated for the species, but the turtle showed serious ocular lesions that completely impaired sight; therefore, it was transferred to the Marine Turtle Research Center, (MTRC) of the Stazione Zoologica Anton Dohrn in Portici, Italy, for further diagnostic investigation and evaluation of possible treatments. Upon arrival at the MTRC, the turtle appeared in good health condition. Physical examination revealed no relevant alterations, except for the eyes. The turtle was normally responsive to external stimuli and had a good appetite, although it was necessary to provide assisted feeding during the first few weeks because of the difficulties in finding the administered food (anchovies, codfish, and mackerel) due to the completely impaired vision. The eyelids were swollen and not separable. On the left eye, only a limited space between ulcerated lid margins was evident, while in the right eye, a wide ulcerated area extended to the lateral canthus of the superior and inferior eyelids, showing fibrovascular tissue. Ocular examination was not possible (Figure 1A,B).

### 2.2. Maintenance Conditions

The turtle was maintained in a recirculating aquatic system composed of three independent tanks (1500 liters each) connected to a dedicated life support system (LSS) equipped with a mechanical sand filter, a moving bed biological filter, a foam fractionator with ozone injection, a UV treatment unit, and a titanium heater for temperature control (22–24 °C). Temperature, pH, redox potential, and salinity were measured daily with a multiparameter probe (model HQ40D, Hach Lange SRL, Lainate, Italy). Nitrates, nitrites, ammonium, and phosphates were measured weekly using a portable spectrophotometer (model DR1900, Hach Lange SRL, Lainate, Italy). Prior to the introduction of the turtle, the system was emptied, cleaned, and disinfected. All surfaces were sprayed with a solution of chlorhexidine 4%, brushed, and rinsed with fresh water. Pipes and filtration units were disinfected through immersion in a sodium hypochlorite solution at a final concentration of 150 ppm and washed twice in fresh water prior to being filled with natural seawater. Spectrophotometric determination of residual chlorine was performed before introducing the turtle into the system. To maintain water quality at safe levels, the water renewal rate was set at 5 to 7% of the total system volume per day.

### 2.3. Bacterial Isolation

Two swabs were collected from the eyelid fissures of both eyes and processed for bacterial and fungal isolation. The samples were enriched in buffered peptone water (Oxoid, UK) and alkaline saline peptone water for 18–24 h at 25 °C, plated onto different enrichment and selective agar plates, and incubated aerobically for 24–72 h at 25 °C. Fungal cultures were negative, whereas several bacterial agents were isolated and subsequently identified by biochemical, phenotypic, and microscopic characteristics. Specifically, *Pseudomonas aeruginosa*, coagulase-negative *Staphylococci*, *Vibrio alginolyticus*, and *Shewanella putrefaciens* were detected in both samples. *P. aeruginosa* was tested for susceptibility to antimicrobials testing using the disk diffusion method, according to the Clinical and Laboratory Standards Institute documents (2014). The antimicrobials tested were amikacin (30 µg; Oxoid), ceftazidime (30 µg; Oxoid), ciprofloxacin (5 µg; Oxoid), gentamicin (10 µg; Oxoid), colistin sulfate (10 µg; Oxoid), and doxycycline (30 µg; Oxoid). The strain was found to be susceptible to amikacin, ceftazidime, ciprofloxacin, and gentamicin and resistant to doxycycline. No intermediate resistance was detected.

### 2.4. Treatment

According to the antibiogram results, ceftazidime (20 mg/kg q 72 h) was administered via intramuscular injection (IM) for four weeks in combination with fluid therapy (50% saline solution 0.9% and 50% Ringer at the final dose of 1% BW SC qd). Topical treatment (gentamicin sulfate ophthalmic ointment 3d-1) was administered for 4 weeks. Systemic administration of Vit. A (500 UI/kg IM q7d) continued for 6 months. The recovery from ocular disease was not satisfactory. The left eye remained closed and did not show substantial changes, while on the right, ulceration of the skin at the level of the lateral canthus of the eyelids widened, and a large dacryolith became evident through the ulcer and was removed (Figure 2A,B).

### 2.5. Diagnostic Imaging

Ultrasonographic (US) examination of both eyes was performed using a general-purpose device equipped with a 12 mHz linear probe (Mylab Class C©-Esaote, Florence, Italy). The US examination revealed a normal right eye, except for small adhesions between conjunctiva and cornea, whereas the left eye was smaller and had wider adhesions between eyelids and cornea. The corneal layers were also thicker and without their normal doubled lined structure. Bilaterally, an anechoic fluid collection was visible in the dorsal conjunctival space. Hyperechoic structures, compatible with salt gland stones, were visible ventrolaterally to both eyes (Figure 3, Figure 4 and Figure 5).

### 2.6. Surgery

Anesthesia was induced with 50 μg/kg medetomidine hydrochloride (Domitor; Pfizer Animal Health, Exton, Pa, USA) and 5 mg/kg ketamine (Ketavet 100; MSD Animal Health), both administered through the intramuscular route, and with 5 mg/kg propofol (Propovet; Zoetis) administered intravenously in the dorsal cervical sinus using a 0.8 × 40 mm needle. Once the jaws were relaxed, a bite block (15·3 cm × 3·2 cm PVC T-piece) was inserted into the turtle’s mouth to hold its jaws open and provide access to the glottis. Curved hemostats (artery forceps) were used gently to open the glottis for the placement of a 6.5 mm endotracheal tube, and anesthesia was maintained with 1.5 to 3% isoflurane (IsoFlo; Zoetis, Parsippany-Troy Hills, NJ, USA).

The eyelid and the periocular skin were prepared for surgery with a topical 5% povidone iodine solution. Surgery was performed under magnification (×4) with a surgical microscope (Shin Nippon OP-2). After separating the eyelids, blunt dissection of the fibrous adhesions between the palpebral and bulbar conjunctiva and the cornea was performed. Then, dissection of the third eyelid was achieved, and it was partially excised, due to the compromise of its margin. The wound was left open for secondary intention healing. In the right eye, the intraocular structures were normal (Figure 6). In the left eye, it was not possible to dissect all the fibrovascular tissue infiltrating the cornea, and the globe appeared smaller and hypotonic. Another large dacryolith was removed from the left orbit.

### 2.7. Postoperative Period and Follow-Up

An ophthalmic ointment containing a combination of tobramycin 0.3% and dexamethasone 0.1% was applied on both eyes TID for 4 weeks after surgery. Tobramycin ophthalmic ointment 0.3% was applied TID on both eyes for an additional 4 weeks. Systemic administration of vit. A continued as described. The day after the surgery, the turtle’s vision, for the right eye, appeared to be normal; it was able to navigate within its tank, avoid objects, and identify and ingest food. Ocular examination confirmed the integrity of the ocular structures in the right eye. Seven days later, the eyelids healed without complications, and lid motility was restored in both eyes. After 14 days, the right eye corneal epithelium had healed and the fluorescence test was negative. The intraocular pressure, measured by rebound tonometry (TonoVet, Icare Finland Oy, Vantaa, Finland; set to the ‘‘dog’’ or ‘‘d’’ manufacturer setting) was 9 mmHg, in physiological limit. Therefore, dexamethasone acetate drops were applied to TID for two weeks to reduce scar tissue formation. There was no recurrence after 12 months of follow-up. In the left eye, there was no recurrence of conjunctivo-palpebral adhesions, and phthisis bulbi persisted. 16 months after the end of treatment, upon restoration of vision to the right eye, the *C. caretta* turtle was tagged with a titanium flipper tag (code SZN 137) and successfully released from the south-western coast of the Tyrrhenian Sea.

### 2.8. Histology and Transmission Electron Microscopy

During surgery, samples of the adherent tissue from the right and left eyes were collected and fixed in 10% buffered formalin for histological evaluation. Briefly, samples were dehydrated through graded alcohols before being embedded in paraffin wax. Sections were cut at 5 micron thickness and stained with hematoxylin–eosin (HE). Specimens from both left and right eyes were irregularly shaped samples of dense fibrovascular tissue covered in some areas by a nonkeratinized stratified squamous epithelium, morphologically compatible with corneal epithelium, and in other areas by stratified columnar epithelium morphologically compatible with conjunctival epithelium. The conjunctival epithelium was diffusely hyperplastic and multifocally lost with erosions and replaced by serocellular crusts composed of lamellated eosinophilic material (keratin), karyorrhectic and cellular debris, and an eosinophilic homogenous material (exudate). The subepithelial connective tissue was involved by a moderate-to-severe multifocal inflammatory infiltrate characterized by lymphocytes and plasma cells (Figure 7). The macroscopic and microscopic findings were consistent with chronic keratoconjunctivitis. For ultrastructural investigations, samples were fixed in 4% glutaraldehyde in 0.1 M phosphate buffer (pH 7.4) for 2–3 h. Then, tissues were postfixed in 1% osmium tetroxide (OsO_4_) in the same buffer for 1 h, dehydrated in a graded alcohol series, and embedded in epoxy resin (EMbed-812, Electron Microscopy Sciences, Hatfield, PA, USA). Ultrathin sections were cut on an EM UC6 ultramicrotome (Leica Microsystems, Shinjuku City, Tokyo, Japan) and collected onto 300-mesh grids. Sections were counterstained with lead citrate and uranyl acetate and examined with a JEOL JEM-1011 transmission electron microscope (JEOL, Tokyo, Japan). Ultrastructurally, no virus-like particles were observed.

### 2.9. Viral Identification

Tissue samples from the ocular lesion during eye surgery were collected for viral identification and immediately frozen at −80 °C. Nucleic acids were extracted after digestion with Proteinase K using Qiagen DNeasy Blood and Tissue Kits (Qiagen Inc., Valencia, CA, USA) according to the manufacturer’s protocol. ChHV-5 PCRs were performed according to the methods described by Rossi et al. [16]. Briefly, 2.5 µL of DNA in a total volume of 25 µL was subjected to PCR amplification using the specific primers for turtle herpesvirus DNA polymerase, GTHV2 (5′-GACACGCAGGCCAAAAAGCGA-3′) and GTHV3 (5′-AGCATCATCCAGGCCCACAA-3′), described by Quackenbush et al. [17]. The standard PCR was performed in 12.625 µL ultrapure water, 2.5 µL buffer solution for PCR (20 mM Tris-HCl pH 8.4 and 50 mM KCl), 4.0 µL dNTP (200 µM of each dNTP), 1.25 µL of each primer (0.4 µM of each primer), 0.75 µL 1.5 mM MgCl2, and 0.125 µL of the enzyme Platinum Taq Polymerase (Invitrogen Life Technologies, Waltham, MA, USA). The sample was denatured at 94 °C for 5 min and then amplified with 35 cycles (94 °C for 30 s, 62 °C for 30 s, 72 °C for 30 s) followed by a 10 min cycle at 72 °C in a thermal cycler. Amplification products were analyzed by 1.5% agarose gel electrophoresis in TBE buffer (89 mM Tris-borate, 2 mM EDTA, pH 8.2) using the ChemiDoc Gel Scanner (Bio-Rad Laboratories, Hercules, CA, USA). None of the tissue samples were positive for ChHV-5 by qualitative PCR.

## 3. Discussion and Conclusions

There are few reported cases of ocular diseases in sea turtles, and they mostly relate to conjunctivitis, keratitis, blepharitis, and corneal injuries [1].

In sea turtles affected by fibropapillomatosis, fibroepithelial tumors are frequently observed on the sclera, cornea, and eyelids [18]. Fibropapilloma tumors differ in appearance, such as color (white, pink, and grey) and texture (smooth to verruciform) [16,19,20,21,22].

Green turtles might also be affected by ChHV-6, which has been suggested to be responsible for lung-eye-trachea disease, characterized by ulceration and accumulation of caseous debris localized to the eye, oropharynx, trachea, and lungs. Another herpesvirus, genetically like ChHV-6 and referred to as loggerhead genital-respiratory herpesvirus, was found in the necropsy of a loggerhead sea turtle exhibiting several lesions, including ulcers along the mucocutaneous junction of the eyelids [23,24,25]. A qualitative PCR test excluded the presence of the chelonid herpesvirus in our case, as all tissue samples were negative. Recently, an atypical case of conjunctivitis due to ChAHV-5 was described in a captive loggerhead turtle with bilateral chemosis, conjunctival hyperemia, and mucoid secretions [3].

In addition, there are very few descriptions of salt gland diseases in wild sea turtles [26]. It has been hypothesized that traumatic factors represent the most common underlying cause of ocular affection in these animals. Hypovitaminosis and bacterial or fungal contamination may also play a significant role [1]. In the case described here, it was not possible to identify the specific etiologic factor responsible for the ocular disease and to describe the exact pathogenesis of symblepharon. The ocular lesions observed could have a multifactorial etiology in which biological and environmental factors could play an important role in the pathogenesis of ocular damage.

Therefore, we hypothesized a possible bacterial proliferation on conjunctival and corneal tissue compromised by chemical burning or hypovitaminosis, or a primitive dysfunction of the salt gland.

The presence of the large dacryoliths in the right and left orbit may be explained by an alteration of the tear fluid dynamic in which the normal secretion of the concentrated sodium chloride solution produced by the lachrymal salt gland in response to increased plasma sodium in sea turtles was inefficient. The lack of elimination of this secretion was probably due to the adhesion of the edges of the upper and lower eyelids.

The pathogenesis of dacriolite formation is still unknown; it was suggested that different mechanisms were involved in the dacryolithogenesis (foreign bodies, trauma, chronic inflammation, bacterial and fungal infections, irritants, etc.) [27].

The bacteria isolated from the eye swabs in this case are considered opportunistic pathogens and have been previously described in association with ocular and salt gland diseases of sea turtles [1,26,28,29]. It is suggested that *Pseudomonas aeruginosa* may cause ulcerative blepharitis and salt gland abscesses in sea turtles [16,18]. Additionally, in two cases, this microorganism caused salt gland adenitis, introduced by contaminated forceps used to remove foreign material from the main excretory duct leading from the posterior orbit [18].

To the authors’ knowledge, this is the first report of symblepharon associated with ankyloblepharon and salt gland dysfunction in *C. caretta*. In the present case, ultrasonography was demonstrated to be the only examination capable of visualizing the eyes and providing a prognosis, as it showed normal right eye structures and dimensions, as previously reported [30].

Surgical correction of the right eye adhesions was successful.

The aim of the surgical correction of symblepharon is to excise the fibrous adhesions between the conjunctiva and cornea and to restore viable epithelial surfaces to the palpebral and bulbar conjunctivae and to the cornea [9]. In our case, because the symblepharon between the bulbar conjunctiva and cornea was not extensive and the remaining freely movable conjunctiva was not large, we removed it without securing it to the limbus and to the conjunctival fornix. It has been reported that adhesions will recur when corneal or conjunctival structures are not covered with epithelium postoperatively [9]. It is also possible to use symblepharon lenses, amniotic membranes, or silicone strips to physically separate the healing conjunctival surfaces and establish and maintain the conjunctival fornix, with these temporary implants held in place by complete temporary tarsorrhaphy [9]. In our case, blunt dissection of adhesions in the right eye was simple and not particularly aggressive, and the covering of the epithelium was not compromised. In addition, the lack of postoperative adhesions suggests that patient management allowed the resolution of the defect. The intraocular pressure, measured by rebound tonometry, was within the physiological range previously described for this species [31]. Excision of the third eyelid margin did not seem to present any deleterious effects on the remaining functional ocular structures.

To prevent recurrence, which occurs in many cases after dissection of conjunctival adhesions and superficial keratectomy, the application of temporary implants such as amniotic membranes (AMs) [32], soft contact lenses (SCLs) [33,34,35], methyl methacrylate corneal protectors [6], gelatin sponges [36], mitomycin C [37], partial limbal stem cell implants [38,39], and platelet-rich fibrin grafts [40] has been described for the treatment of symblepharon in human and veterinary ophthalmology. When adhesions continue beyond the limbus into the conjunctiva, these too are severed [6]. In this case report, the surgical treatment alone was decisive, probably because the corneal surface beyond the limbus was not affected, resulting in limbal stem cell deficiency.

Given the importance of vision to the survival of sea turtles in the wild, symblepharon should be included among sea turtle eye diseases, and a thorough examination of the eye should be performed to improve prognosis and reintroduce these endangered species to their natural habitat successfully.

## Figures and Tables

**Figure 1 vetsci-09-00281-f001:**
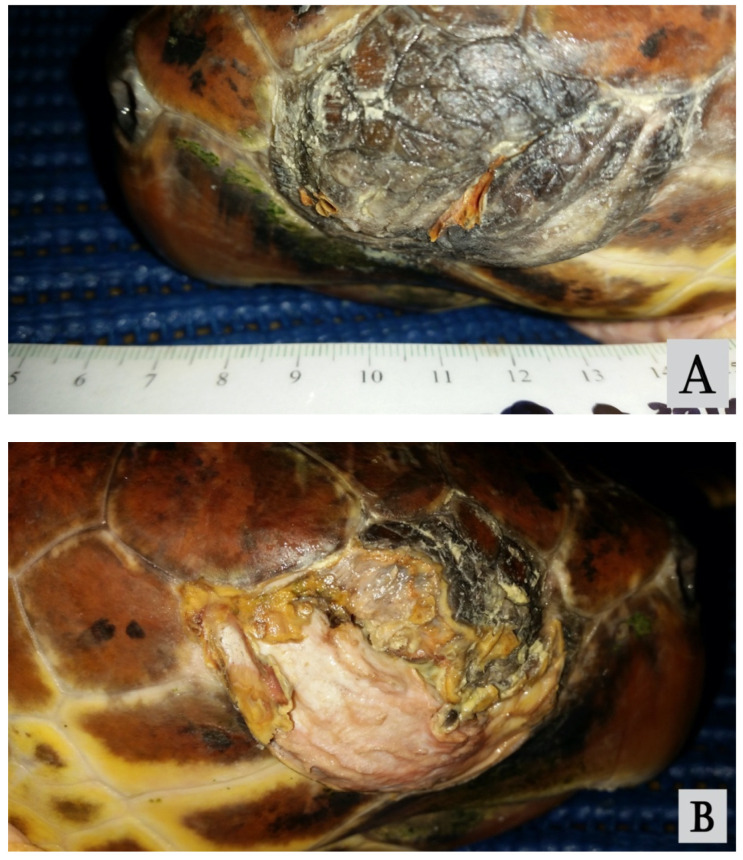
The left (**A**) and right (**B**) eyes at initial presentation, showing swollen, ulcered, and unseparable eyelids.

**Figure 2 vetsci-09-00281-f002:**
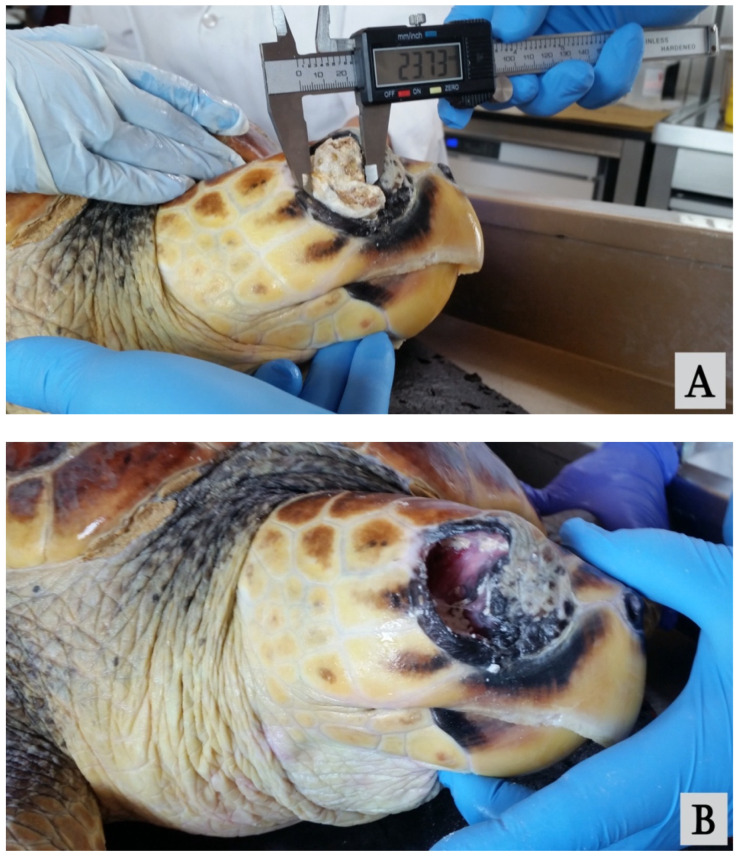
Dacryolith protruding through the eyelid ulcer (**A**) and free orbit space posterior to the ocular globe after removal (**B**).

**Figure 3 vetsci-09-00281-f003:**
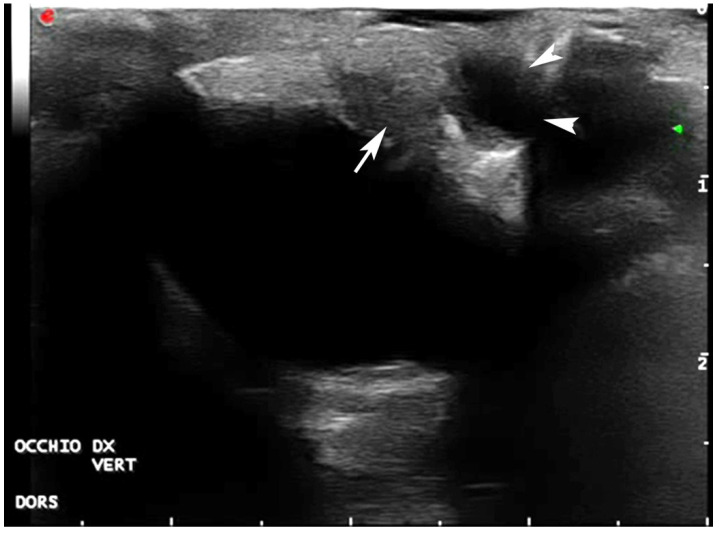
Ultrasonography of the right eye, vertical lateral-parasagittal scan: adhesion between the eyelid and cornea is seen (arrow). A fluid collection in the dorsal conjunctival space is also present (arrowheads).

**Figure 4 vetsci-09-00281-f004:**
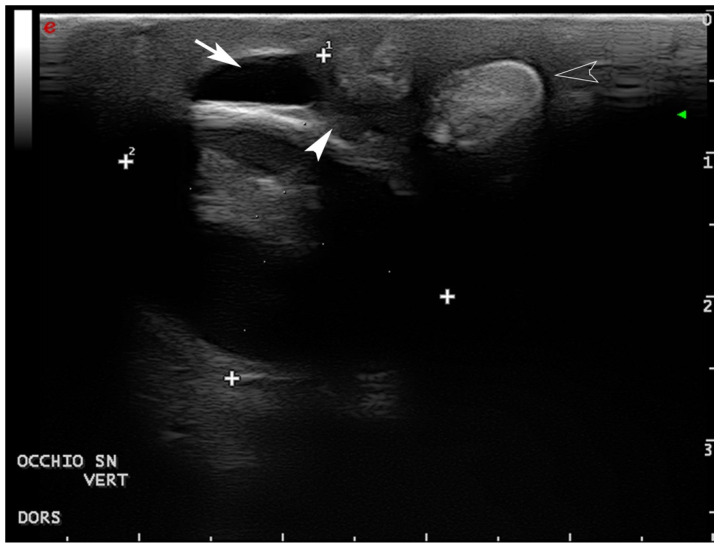
Ultrasonography of the left eye, vertical lateral-parasagittal scan: a fluid collection in the dorsal conjunctival space is present (arrow) delimitated ventrally from adhesion between the eyelid and cornea (arrowhead). More ventrally, a salt dacryolith is visible at the gland level (empty arrowhead). (Plus are electronic calipers to evaluate the eye depth ^1^ and height ^2^).

**Figure 5 vetsci-09-00281-f005:**
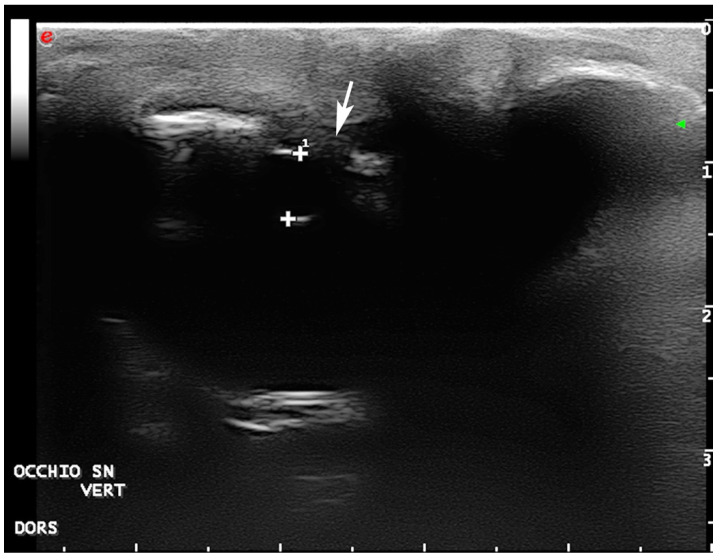
Ultrasonography of the left eye, vertical scan: a normal lens is visible (4.6 mm, the space between the electronic calipers “+”) but not the anterior chamber; the cornea is thickened and without the normal doubled layer structure (arrow).

**Figure 6 vetsci-09-00281-f006:**
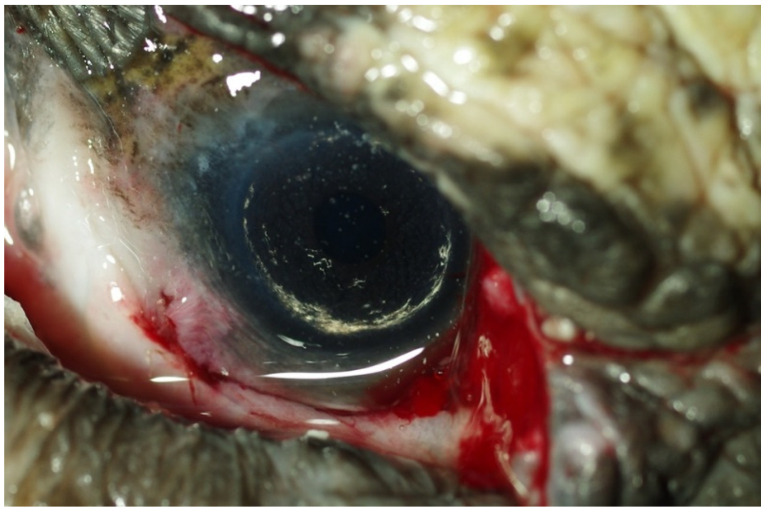
Post-surgery view of the right eye. After dissection of nictitans and adhesions between the palpebral and bulbar conjunctiva and cornea, the eyeball appeared normal.

**Figure 7 vetsci-09-00281-f007:**
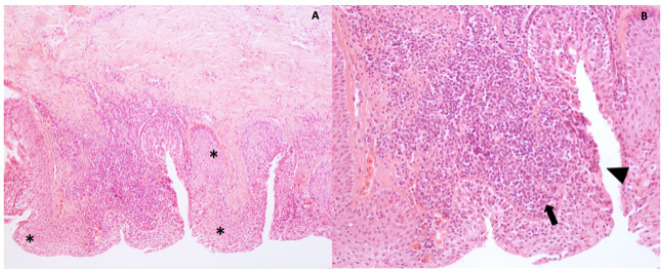
(**A**,**B**) Tissue from the left eye. A dense fibrovascular tissue covered by an epithelium diffusely hyperplastic (hyperplasia, asterisks) and multifocally lost (erosion, arrowhead). The subepithelial connective tissue is focally infiltrated by mononuclear inflammatory cells (arrow). Hematoxylin and eosin stain, 10× and 20×.

## Data Availability

The data supporting our findings is contained within the article.

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
