# Peer review of "Symblepharon, Ankyloblepharon, and Salt Gland Dysfunction in a Loggerhead Sea Turtle (Caretta caretta)"

_vetsci, 2022, doi:10.3390/vetsci9060281_

Round 1
Reviewer 1 Report
Type of manuscript: Case Report
Title: Symblepharon, Ankyloblepharon, and Salt Gland Dysfunction in a
Loggerhead Sea Turtle (Caretta Caretta)
Journal: Veterinary Sciences
Reviewers comments to authors
The paper is interesting and focuses in na unsusual clinical case in a sea turtle. It is generally well written, with a good balance between sections. The iconography is very interesting, English language should be revised.
Oberall I believe that a major revision should be done before it is suitable for publication.
Abstract section
- line 17 «fixity»? Rephrase
- line 19 - «stranded»? rephrase
- line 21 – add surgical before dissection
- line 25 – ont only the eyelids had ot heal, please be more accurate
- line 26 – by «released» do you mean into the ocean?
- Line 28 – a Caretta caretta turtle (italic)
Introduction
Please add a paragraph on most common ocular diseases in sea turtles.
Case presentation
Line 66 – ocular lesions
Line 77 – ocular examination ,,, not possible to perform.
Line 133 – that were also thicker and without its normal doubled lined
Structure- What was thicker? The adesions, the eyelids or the cornea layers?
Discussion
Line 273 - The surgical correction was successful for the right eye. Rephrase: Surgical correction of the right eye adhesions was sucessful
Line 287 – defect and not effect
Line 287 - «was, in physiological range previously described in this species.» Rephrase: «was within the physiological range previously described for this species.»
What about the use of Mitomicin C to avoid reepithelization of the cornea?
Could you provide possible explanations for the presence of the extense dacryolith in the right orbit?
Author Response
REFEREE 1
The paper is interesting and focuses in an unusual clinical case in a sea turtle. It is generally well written, with a good balance between sections. The iconography is very interesting, English language should be revised.
Dear Referee 1, thank you for your appreciation.
Thanks to your suggestions, we revised the manuscript with a native English speaker.
Oberall I believe that a major revision should be done before it is suitable for publication.
Abstract section
line 17 «fixity»? Thank you for advising. In the revision, we did rewrite the sentence: “adhesion of the edges of upper and lower eyelids”.
line 19 - «stranded»? Rephrase: “The loggerhead presented both eyelids swollen, ulcerated, and not separable when it had been rescued”.
line 21 – add surgical before dissection. In the revision, we did rewrite the sentence
line 25 – ont only the eyelids had ot heal, please be more accurate. We did rewrite the sentence: “the eyelids of both eyes and the corneal epithelium of right eye healed”.
line 26 – by «released» do you mean into the ocean? We added: “on the southern Tyrrhenian coast in the western Mediterranean Sea”
Line 28 – a Caretta caretta turtle (italic) “Caretta caretta turtle”
Introduction
Please add a paragraph on most common ocular diseases in sea turtles.
Most common ocular abnormalities in sea turtles are blepharitis, keratitis, and conjunctivitis; fibropapillomatosis, which is thought to be caused by a virus in the alphaherpesvirus family is also frequent (IÅŸler CT, AltuÄŸ M, CantekÄ°n Z, et al. Evaluation of the eye diseases seen in Loggerhead Sea turtle (Caretta caretta). Rev Med Vet. 2014; 165:258-262.) (Flint M, Limpus CJ, Patterson-Kane JC, et al. Corneal fibropapillomatosis in green sea turtles (Chelonia mydas) in Australia. J Comp Pathol 2010; 142:341–346.). An atypical case of conjunctivitis by ChAHV-5, was recently described in a captive loggerhead turtle (Arianne P. Oriá, Danielle N. Silva, Ana Cláudia Raposo, Alessandra Estrela- Lima, Thaís T. Pires, Marco A. Gattamorta, Roberta R. Zamana, Eliana R. Matushima, Ron Ofri. Atypical ocular Chelonoid herpesvirus manifestations in a captive Loggerhead turtle (Caretta caretta) Veterinary Ophthalmology. 2021;24:97–102)
Case presentation
Line 66 – ocular lesions
Line 77 – ocular examination ,,, not possible to perform.
Thank you. In the revision, we did rewrite the sentence
Line 133 – that were also thicker and without its normal doubled lined structure- What was thicker? The adesions, the eyelids or the cornea layers? Rephrase: “The cornea layers”
Discussion
Line 273 - The surgical correction was successful for the right eye. Rephrase: Surgical correction of the right eye adhesions was sucessful
Line 287 – defect and not effect Thank you for pointing this out. We changed this word.
Line 287 - «was, in physiological range previously described in this species.» Rephrase: «was within the physiological range previously described for this species.» Thank you. We changed this wording with your suggestion.
What about the use of Mitomicin C to avoid reepithelization of the cornea?
To prevent recurrence that in many cases evolves after dissection of conjunctival adhesions and superficial cheratectomy, application of temporary implants such as amniotic membranes (AMs) [Patel AP, Satani DR, Singh S, Desai S. Application of amniotic membrane transplantation in cases of symblepharon. J Indian Med Assoc 2012;110(6):388-389.], soft contact lenses (SCLs) (Daglioglu MC, Coskun M, Ilhan N, Tuzcu EA, Ilhan O, Keskin U, et al. The effects of soft contact lens use on cornea and patient's recovery after autograft pterygium surgery. Cont Lens Anterior Eye. 2014;37(3):175-177. Kaufman HE, Thomas EL. Prevention and treatment of symblepharon. Am J Ophthalmol. 1979;88(3 Pt 1):419-423; Youngsam Kim, Seonmi Kang, Kangmoon Seo. Application of superficial keratectomy and soft contact lens for the treatment of symblepharon in a cat: a case report. J Vet Sci. 2021 Mar;22(2) 1-5 ], methyl methacrylate corneal protectors [Gelatt KN, Brooks DE. Surgical procedures for the conjunctiva and the nictitating membrane. In: Gelatt KN,Gelatt JP, editors. Veterinary Ophthalmic Surgery. 1st ed. Maryland Height: Elsevire Saunders; 2011, 157-190.], gelatin sponges [Yamada M, Sano Y, Watanabe A, Mashima Y. Preventing symblepharon formation with a gelatin sponge in the eye of a patient with an alkali burn. Am J Ophthalmol. 1997;123(4):552-554.], mitomycin C [Rodriguez JA, Ferrari C, Hernández GA. Intraoperative application of topical mitomycin C 0.05% for pterygium surgery. Bol Asoc Med P R 2004;96(2):100-102.], partial limbal stem cell implants [Shi W, Wang T, Gao H, Xie L. Management of severe ocular burns with symblepharon. Graefes Arch Clin Exp Ophthalmol. 2009;247(1):101-106. Hille K, Makuch D, Wilske J, Ruprecht KW. The effectiveness of limbus epithelium transplantation. Ophthalmologe. 2002;99(7):575-579.] and platelet-rich fibrin grafts (Ning Yang, Yiqiao Xing, Qiuya Zhao, Siyu Zeng, Juan Yang, Lei Du. Application of platelet-rich fibrin grafts following pterygium excision Int J Clin Pract. 2021;75:e14560.1-7) has been described for the treatment of symblepharon in human and veterinary ophthalmology. If the adhesions continue into the conjunctiva beyond the limbus, these too are severed (Gelatt KN, Brooks DE. Surgical procedures for the conjunctiva and the nictitating membrane. In: Gelatt KN,Gelatt JP, editors. Veterinary Ophthalmic Surgery. 1st ed. Maryland Height: Elsevire Saunders; 2011, 157-190). In this case report, the surgical treatment alone was decisive, probably for the lack of involvement of corneal surface beyond the limbus resulting in limbal stem cell deficiency.
Could you provide possible explanations for the presence of the extense dacryolith in the right orbit?
The presence of the large dacryoliths in the right and left orbit may be explained by an alteration of tear fluid dynamic in which the normal secretion of the concentrated sodium chloride solution produced by the lachrymal salt gland in response to increased plasma sodium in sea turtles was inefficient.
We appreciate all your insightful comments. Thanks for taking the time and energy to help us improving our manuscript.
Best regards.
The authors
Reviewer 2 Report
The paper is interesting, even for a non-veterinarian.
Here is a list a minor changes required:
In title, the species name "Caretta" must not be with a cap letter.
Italicized binomial names in title (line 3), in abstract (line 28), in 2.3 Bacterial isolation lines 106 and 107, as well as in the references 3, 4, 5, 15 (also change Caretta Caretta to Caretta Caretta), 19, 24 and in 28 change Caretta Caretta to Caretta caretta.
Correct author name in ref 3: Brandão. Remove the ` after histopathological in this reference.
In the introduction, a schema of the marine turtle eye structures is necessary to follow the descriptions.
The authors searched for ChHV-5 virus but do not searched for ChHV-6. Authors should discuss that point.
The authors do not indicate the size and weight of the individual when it was released. Does this individual was tagged using tag or PIT ?
Author Response
REFEREE 2
The paper is interesting, even for a non-veterinarian.
Dear Referee 2, thank you for your appreciation.
Here is a list a minor changes required:
In title, the species name "Caretta" must not be with a cap letter.
Italicized binomial names in title (line 3), in abstract (line 28), in 2.3 Bacterial isolation lines 106 and 107, as well as in the references 3, 4, 5, 15 (also change Caretta Caretta to Caretta Caretta), 19, 24 and in 28 change Caretta Caretta to Caretta caretta.
Correct author name in ref 3: Brandão. Remove the ` after histopathological in this reference.
Thank you for your suggestion. In the revision, we did rewrite the sentences.
In the introduction, a schema of the marine turtle eye structures is necessary to follow the descriptions.
The eye in the Loggerhead Sea turtle has three keratinized eyelids, two mobile, dorsal and ventral eyelids, and a nonmobile medial lid called the “secondary lid” [4, 5]. Ventral lid and nictitating membrane are continuous with the conjunctiva. Salt gland is the largest organ in the head of sea turtles. There is also an Harderian gland located anterior to the eye. Each salt gland envelops the caudal orbit, extends caudally and medially to the eye and drains in a duct that opens in the caudo-dorsal conjunctiva [4, 5]. This organ enables sea turtles to maintain osmotic and ionic homeostasis in a hypersaline environmental medium by secreting a concentrated sodium chloride solution in response to increased plasma sodium [6, 7, 8].
The authors searched for ChHV-5 virus but do not searched for ChHV-6. Authors should discuss that point.
We chose to search only for ChAHV-5 because this virus most often has a clinical manifestation in the eye. In addition, ChAHV-5 infection in sea turtles causes a lesion like fibropapillomatosis (Oriá et al., 2021).
Furthermore, most investigations interested in sequence data from ChHV-5 rely on samples sourced from tumors or the skin immediately adjacent to tumors (Ene et al., 2005; Patricio et al., 2012). The high concentration of viral DNA in these tissues simplifies the process of amplification, making them attractive targets (Lawrance et al., 2018).
Since the turtle was in a good general condition and showed no respiratory symptoms, we focused our investigations only on this pathogen. Finally, ChAHV-5 genome has already been isolated in C. caretta specimens affected by eye lesions.
The authors do not indicate the size and weight of the individual when it was released. Does this individual was tagged using tag or PIT ? The C. caretta turtle was tagged with a titanium flipper tag (code SZN 137) and successfully released from the southwestern coast of the Tyrrhenian Sea.
We appreciate all your insightful comments. Thanks for taking the time and energy to help us improving our manuscript.
Best regards.
The authors